# The Significance of Longitudinal Psoas Muscle Loss in Predicting the Maintenance Efficacy of Durvalumab Treatment Following Concurrent Chemoradiotherapy in Patients with Non-Small Cell Lung Cancer: A Retrospective Study

**DOI:** 10.3390/cancers16173037

**Published:** 2024-08-30

**Authors:** Haruka Kuno, Naoya Nishioka, Tadaaki Yamada, Yusuke Kunimatsu, Akihiro Yoshimura, Soichi Hirai, Shun Futamura, Taiki Masui, Masashi Egami, Yusuke Chihara, Koichi Takayama

**Affiliations:** 1Department of Pulmonary Medicine, Graduate School of Medical Science, Kyoto Prefectural University of Medicine, Kyoto 602-0841, Japan; kuno@koto.kpu-m.ac.jp (H.K.); g4h4n93w@koto.kpu-m.ac.jp (N.N.); ky9202@koto.kpu-m.ac.jp (Y.K.); hirasoh9@koto.kpu-m.ac.jp (S.H.); masui@koto.kpu-m.ac.jp (T.M.); megami@koto.kpu-m.ac.jp (M.E.); takayama@koto.kpu-m.ac.jp (K.T.); 2Department of Respiratory Medicine, Japanese Red Cross Kyoto Daini Hospital, Kyoto 602-8026, Japan; aki-y@koto.kpu-m.ac.jp; 3Department of Medical Oncology, Fukuchiyama City Hospital, Fukuchiyama 620-8505, Japan; shunf@koto.kpu-m.ac.jp; 4Department of Respiratory Medicine, Uji-Tokushukai Medical Center, Uji 611-0041, Japan; c1981311@koto.kpu-m.ac.jp

**Keywords:** muscle change, concurrent chemoradiotherapy, durvalumab, sarcopenia, immunotherapy

## Abstract

**Simple Summary:**

Sarcopenia refers to a progressive and systemic reduction in skeletal muscle mass and strength and is a primary component of cancer cachexia. This retrospective study examined the impact of the rate of muscle loss between two time points during the pretreatment period with concurrent chemoradiotherapy on the clinical outcomes during subsequent durvalumab maintenance treatment. As the results showed, patients who experienced psoas muscle loss during chemoradiotherapy had a shorter progression-free survival and experienced reduced effectiveness of durvalumab treatment. This study highlights the importance of monitoring psoas muscle changes during chemoradiotherapy to predict the success of subsequent durvalumab therapy better in non-small cell lung cancer patients.

**Abstract:**

Sarcopenia assessed at a single time point is associated with the efficacy of immunotherapy, and we hypothesized that longitudinal changes in muscle mass may also be important. This retrospective study included patients with non-small cell lung cancer (NSCLC) who received durvalumab treatment after concurrent chemoradiotherapy (CCRT) between January 2017 and April 2023. Muscle loss and sarcopenia were assessed based on the lumbar skeletal muscle area. Patients with a decrease in muscle area of 10% or more during CCRT were categorized into the muscle loss group, while those with a decrease of less than 10% were categorized into the muscle maintenance group. We evaluated the relationship between muscle changes during CCRT and the efficacy of durvalumab treatment. Among the 98 patients, the muscle maintenance group had a significantly longer PFS of durvalumab treatment compared to the muscle loss group (29.2 months [95% confidence interval (CI): 17.2—not reached] versus 11.3 months [95% CI: 7.6–22.3]; *p* = 0.008). The multivariable analysis confirmed that muscle change was a significant predictor of a superior PFS (HR: 0.47 [95% CI: 0.25–0.90]; the *p*-value was less than 0.05). In contrast, the OS between the groups did not differ significantly (not reached [95% CI: 21.8 months—not reached] and 36.6 months [95% CI: 26.9—not reached]; *p* = 0.49). Longitudinal muscle changes during CCRT are a predictor of durvalumab’s efficacy in patients with NSCLC after CCRT.

## 1. Introduction

Since their approval for the treatment of advanced lung cancer, immune checkpoint inhibitors (ICIs) have demonstrated durable efficacy compared to conventional chemotherapy in patients with non-small cell lung cancer (NSCLC), although ICIs are ineffective in certain patients [1,2,3,4,5]. Therefore, identifying the predictors of treatment response to ICI therapy is a critical clinical task. For patients with unresectable stage III NSCLC, concurrent chemoradiotherapy (CCRT) followed by ICI treatment is the standard of care [6]. Currently, programmed cell death ligand 1 (PD-L1) expression in tumors is universally used as a major predictor of the response to immunotherapy. However, this is not a sufficiently comprehensive factor [7,8,9]. Tumor mutational burden (TMB) has been suggested as another potential predictor. TMB refers to the total number of mutations per coding area of a tumor genome, and it has been associated with an increased neoantigen load and a better response to ICIs [10]. However, determining the cutoff for high TMB levels remains problematic. Furthermore, even among patients with a high TMB, only 29% responded to treatment, highlighting the limitations of TMB as a biomarker for ICI therapy [11,12]. Another factor is tumor aneuploidy, a condition characterized by abnormal chromosome copy numbers that is common in most cancers [13]. Aneuploidy plays a crucial role in the growth and survival of cancer cells and may promote tumor progression by suppressing antitumor immunity. Additionally, tumors with high levels of aneuploidy often have a poor prognosis in response to ICI therapy [14,15]. However, conclusive evidence supporting their efficacy remains lacking.

Sarcopenia refers to a progressive and systemic reduction in skeletal muscle mass and strength and is a primary component of cancer cachexia [16]. Sarcopenia is associated with a poor prognosis in solid cancers, including lung cancer [17]; approximately half of patients with stage III-IV NSCLC are affected by sarcopenia [18]. In patients with NSCLC with sarcopenia, a shorter progression-free survival (PFS) and overall survival (OS) have been reported compared to those in patients with NSCLC without sarcopenia [19,20,21,22,23]. Although accumulating evidence suggests that sarcopenia at a single time point is associated with the response to ICI therapy, assessing the rate of muscle loss at multiple time points from cancer onset to treatment initiation dynamically reflects sarcopenia independent of preexisting changes in body size and muscle mass at cancer diagnosis and can provide a more precise assessment. Recently, sarcopenia has been diagnosed based on skeletal muscle mass measurements using computed tomography (CT) images [24,25].

This study examines the impact of the rate of muscle loss between two time points during the pretreatment period with CCRT on the PFS and OS during subsequent ICI maintenance treatments. We also investigate whether the progression of sarcopenia at a given time point in this cohort could predict the treatment response to ICIs.

## 2. Materials and Methods

### 2.1. The Study Population

The study population included patients with NSCLC who received durvalumab treatment for at least one cycle after CCRT between January 2017 and April 2023 at four medical institutions. The reasons for exclusion were as follows: (a) the CT images obtained before chemoradiotherapy did not include images extending to the level of the third lumbar vertebra (L3) and (b) the CT images obtained before durvalumab administration did not include images extending to the L3 level. The institutional review board of our hospital approved this study and waived the requirement for written informed consent due to the retrospective nature of the study.

### 2.2. Data Collection

The following data were collected from the medical records of eligible patients: age, sex, Eastern Cooperative Oncology Group performance status (ECOG-PS) just before the initiation of chemoradiotherapy and durvalumab administration, stage, histology, driver oncogenes, PD-L1 tumor proportion score (TPS), type of chemotherapy, date of starting chemoradiotherapy and durvalumab administration, and the presence of grade 3 or higher gastrointestinal adverse events/anorexia/radiation esophagitis during CCRT according to the Common Terminology Criteria for Adverse Events (CTCAE version 5.0). Height and weight were measured within a week before chemoradiotherapy and durvalumab administration, and the body mass index (BMI) was calculated from these data. Cachexia was defined as weight loss >5% during chemoradiotherapy or >2% in patients with a BMI < 20 kg/m^2^. We obtained imaging data from chest and thoracoabdominal CT images obtained within 2 months before chemoradiotherapy and 1 month before durvalumab administration. These scans were used to collect data on the cross-sectional area of the lumbar skeletal muscle at L3 before chemoradiotherapy and durvalumab administration.

### 2.3. Evaluation of Muscle Quantity and Sarcopenia

Computed tomography scans with a slice thickness of 1–5 mm taken before chemoradiotherapy and durvalumab administration were used to identify the lumbar vertebral skeletal muscles, using a threshold of −29 to +150 Hounsfield units (HU). In this cohort, the cutoff value for muscle loss determined by ROC curve analysis of the durvalumab completion rates was 6.6% (area under the curve: 0.61; 95% confidence interval: 0.49–0.72) (Appendix A). Additionally, based on previous studies, we classified NSCLC patients with a reduction of 10% or more in the cross-sectional area of their bilateral psoas muscles at the L3 level as belonging to the muscle loss group and those with a reduction of less than 10% as belonging to the muscle maintenance group in this study [26,27]. In addition to these criteria, the cross-sectional area of the psoas muscle at L3 was measured in the CT slices, and the psoas muscle mass index (PMI) was calculated by dividing this value by the square of the patient’s height to evaluate sarcopenia. The cutoff values for sarcopenia were set at 6.36 cm^2^/m^2^ for men and 3.92 cm^2^/m^2^ for women [25].

### 2.4. Statistical Analyses

The primary endpoint was PFS, which was defined as the duration from the initial administration of durvalumab to the first occurrence of disease progression according to the Response Evaluation Criteria in Solid Tumors (RECIST) criteria or death from any cause, whichever occurred first. Overall survival was defined as the period from the initial administration of durvalumab to the day of death from any cause or to the last confirmed date of survival. The survival analysis was conducted with the cutoff date for the entire cohort set as 31 May 2023. Progression-free survival and OS were calculated using the Kaplan–Meier method. Survival curves for the muscle loss and maintenance groups were compared using a log-rank test. Potential predictors were assessed using Cox proportional hazard models for PFS. Continuous variables are presented as medians (ranges or 95% confidence intervals), and comparisons between two groups were conducted using the Mann–Whitney U test. Nominal variables are presented as numbers (*n*) or percentages (%), and comparisons between two groups were performed using Fisher’s exact test. Statistical significance was set at *p* < 0.05. All the analyses were performed using EZR (Saitama Medical Center, Jichi Medical University, Saitama, Japan), which is a graphical user interface for R (R Foundation for Statistical Computing, Vienna, Austria). More precisely, it is a modified version of the R commander (version 2.9-1) designed to add statistical functions frequently used in biostatistics.

## 3. Results

### 3.1. Patient Characteristics and Treatment

Of 131 consecutive patients with NSCLC who initially received durvalumab for at least one cycle after chemoradiotherapy between January 2017 and April 2023 in four medical institutions, 98 patients were enrolled (Figure 1). The patient characteristics are provided in Table 1. The median age of the patients was 72 years (range: 49–87 years); a total of 73 (74.5%) patients were men, all with an ECOG-PS score of 0–1. The majority of the population comprised patients with stage III disease (89.7%), and 45 (45.9%) had adenocarcinomas, of whom 14 (14.3%) had driver oncogenes. In addition, the TPS was available for 86 (87.8%) patients; the TPS was <50% in 47 (46.9%) patients. The median BMI was 21.9 (range: 15.6–30.5) kg/m^2^ pre-chemoradiotherapy and 21.2 (range: 15.2–28.7) kg/m^2^ post-chemoradiotherapy. Further, 82 (83.7%) and 87 (88.8%) patients had sarcopenia before chemoradiotherapy and durvalumab administration, respectively. In addition, approximately half of the patients (54.1%) had cancer cachexia before chemoradiotherapy. The median interval from the initiation of chemoradiotherapy to the administration of durvalumab was 15.0 (range: 1.0–49.0) days. The completion rate for durvalumab treatment was 50% (49 cases). During the course of CCRT, the most common hematological toxicities were leukopenia (49 patients, 50%), neutropenia (43 patients, 43.9%), anemia (25 patients, 25.5%), and thrombocytopenia (20 patients, 20.4%). Toxicities of grade 3 or higher included leukopenia (40 patients, 40.8%), neutropenia (35 patients, 35.7%), and anemia (3 patients, 3.1%) and no severe thrombocytopenia cases. Febrile neutropenia occurred in six patients (6.1%). All grades of non-hematological toxicity included radiation pneumonitis (42 patients, 42.9%), esophagitis (41 patients, 41.8%), anorexia (34 patients, 34.7%), and hypoalbuminemia (18 patients, 18.4%). Non-hematologic toxicities of grade 3 or higher included esophagitis (14 patients, 14.3%), anorexia (11 patients, 11.2%), and radiation pneumonitis (2 patients, 2.0%) and no severe hypoalbuminemia cases.

The muscle maintenance (*n* = 62) and muscle loss groups (*n* = 36) showed no differences, except for those in the pre-chemotherapy median BMI (21.5 vs. 23.0 kg/m^2^, *p* < 0.05), the pre-chemoradiotherapy PMI in men (4.68 vs. 5.25 cm^2^/m^2^, *p* < 0.05), and the median duration from the start of chemoradiotherapy to the initiation of ICI treatment (21 days vs. 14 days, *p* < 0.05).

### 3.2. Impact of Sarcopenia on Durvalumab’s Efficacy at Different Time Points in Patients with NSCLC

We evaluated the effects of sarcopenia on the therapeutic efficacy of durvalumab in patients with NSCLC before chemoradiotherapy and durvalumab administration. The patients were divided into two groups based on the presence or absence of sarcopenia at two different time points: before chemoradiotherapy and before initiating durvalumab treatment. In the study population, 82 (83.6%) and 87 (88.8%) patients were diagnosed with sarcopenia before chemoradiotherapy and durvalumab administration, respectively. At both evaluation points, patients with sarcopenia tended to have shorter durvalumab efficacy durations, but no statistically significant differences were observed in PFS or OS based on the presence or absence of sarcopenia (Appendix A).

### 3.3. Impact of Longitudinal Muscle Loss on Clinical Outcomes of Durvalumab Treatment

As of the data cutoff date of 31 May 2023, with a median follow-up of 19.1 months (range: 0.6–57.4), 47/98 patients (48%) had experienced progressive disease. The muscle maintenance group had a longer PFS with durvalumab administration than the muscle loss group (29.2 months [95% CI: 17.2—not reached] versus 11.3 months [95% CI: 7.6–22.3], *p* = 0.008, Figure 2A). The multivariable analysis adjusted for age, ECOG PS score, and PD-L1 expression revealed that muscle maintenance was significantly associated with a superior PFS (hazard ratio: 0.47 [95% CI: 0.25–0.90], *p* < 0.05, Table 2). Of the 98 patients, 31 (31.6%) died before the cutoff date. The OS did not significantly differ between the two groups; respectively, the median OS was not reached (95% CI: 21.8 months not reached) and 36.6 months (95% CI: 26.9 months not reached) (*p* = 0.49, Figure 2B).

### 3.4. Combined Analysis of Longitudinal Muscle Loss and Single Time Point Sarcopenia in Predicting Durvalumab’s Efficacy

We categorized patients with NSCLC into four groups based on their longitudinal muscle loss combined with the presence or absence of sarcopenia at two specific time points: before chemoradiotherapy and before durvalumab administration. For the analysis based on sarcopenia status before chemoradiotherapy and muscle loss, the groups were defined as follows: non-sarcopenia with muscle maintenance; sarcopenia with muscle maintenance; non-sarcopenia with muscle loss; and sarcopenia with muscle loss (8 patients, PFS: NA [95% CI: NA–NA]; 54 patients, PFS: 21.9 months [95% CI: 15.5–NA]; 8 patients, PFS: 13.4 months [95% CI: 2.30–NA]; 28 patients, PFS: 11.1 months [95% CI: 5.49–22.31], respectively, *p* = 0.051) (Appendix A). For the evaluation of the association between sarcopenia status before durvalumab administration and muscle loss, the groups were defined as follows: non-sarcopenia with muscle maintenance; sarcopenia with muscle maintenance; non-sarcopenia with muscle loss; and sarcopenia with muscle loss (8 patients, PFS: NA [95% CI: NA–NA]; 54 patients, PFS: 21.9 months [95% CI: 15.5–NA]; 3 patients, PFS: 9.0 months [95% CI: 2.3–NA]; 33 patients, PFS: 11.3 months [95% CI: 7.6–22.3], respectively, *p* = 0.14) (Appendix A). Although no statistically significant differences were observed at any time point, there was a trend toward a shorter PFS with ICIs in patients with sarcopenia or a reduced muscle mass before chemoradiotherapy. Both conditions resulted in poor PFS outcomes. Conversely, before durvalumab treatment, a consistent trend of a shorter PFS with ICIs was observed in patients with a reduced muscle mass, regardless of the presence of sarcopenia (Appendix A).

## 4. Discussion

Muscle loss during chemoradiotherapy adversely affected the therapeutic efficacy of durvalumab maintenance therapy following CCRT in patients with NSCLC. However, the presence of sarcopenia at a single point, either before chemoradiotherapy or durvalumab administration, was not significantly associated with the therapeutic maintenance effect of durvalumab treatment, suggesting that longitudinal muscle loss may be a better predictor of treatment response to immunotherapy than a single-point assessment of skeletal muscle. Furthermore, on comparing the four groups based on longitudinal muscle loss and the presence or absence of sarcopenia, the muscle loss group consistently demonstrated a lower efficacy of durvalumab, regardless of the sarcopenia status before durvalumab administration.

The therapeutic effect of PD-1/PD-L1 inhibitors is associated with sarcopenia in patients with NSCLC [19,20,21,22,28,29,30,31]. In these previous reports, muscle loss (sarcopenia) was defined solely by the muscle mass observed in CT scans at a single point before treatment and suggested that the therapeutic effect of PD-1/PD-L1 inhibitors was significantly lower in the sarcopenia group [19,21,26]. We also evaluated the efficacy of durvalumab based on the presence or absence of sarcopenia before chemoradiotherapy and durvalumab administration. Although no significant differences were found, patients with sarcopenia tended to have a shorter PFS. However, when comparing the four groups based on longitudinal muscle loss and sarcopenia status before durvalumab administration, patients with muscle loss had a shorter PFS regardless of sarcopenia status. Therefore, in a cohort of patients with NSCLC receiving durvalumab after CCRT, longitudinal muscle mass loss may be a more sensitive predictor of treatment efficacy than sarcopenia before durvalumab administration. Roch et al. and Loosen et al. have linked changes in muscle mass over time to the therapeutic effects of ICIs [22,32]. These findings support our finding that longitudinal muscle loss is associated with the therapeutic effects of ICIs.

There are several hypotheses explaining why longitudinal muscle loss before ICI administration may adversely affect ICIs’ effectiveness. One hypothesis is that muscle loss serves as a marker of chronic inflammation [33,34]. In chronic inflammation caused by cancer cachexia, inflammatory cytokines such as TNF-α and IL-6 promote muscle loss and tumor growth [35,36]. Chronic inflammation promotes tumor progression by suppressing CD8+ T-cell infiltration through inflammatory cytokines/mediators or the induction of regulatory T cells and myeloid-derived suppressor cells (MDSCs), which are immunosuppressive cells [37,38]. Therefore, muscle loss may serve as a biomarker of chronic inflammation, and the insufficient effectiveness of ICIs may be attributed to altered tumor microcirculation caused by chronic inflammation. Longitudinal changes in muscle quantity, which allow for more precise assessments, may be more sensitive to the effects of chronic inflammation than single-point measurements. A second hypothesis is that exercise in the muscle maintenance group during chemoradiotherapy may have boosted muscle strength and antitumor immunity, thereby enhancing the ICI treatment’s efficacy. Unfortunately, we were unable to collect patient data on rehabilitation and exercise during the radiotherapy period in this study. However, in general, the muscle-maintaining group tended to maintain their muscle strength better than the muscle loss group [39,40]. In addition, exercise is beneficial for patients with lung cancer, as it promotes muscle growth, maintains muscle strength, increases appetite, and prevents weight loss [39,40]. In animal models, exercise suppresses tumor growth by altering the tumor microenvironment and increasing the levels of NK and CD8+ T cells while also reducing the growth of regulatory T cells and MDSCs [41,42]. Additionally, exercise reduces inflammation and stimulates the skeletal muscle cells to produce immune-regulating cytokines (myokines) [43,44,45,46]. Based on these results, we speculate that exercise may maintain and enhance the therapeutic effects of ICIs by suppressing inflammation and altering the tumor microenvironment. Although adverse events from prior treatments may have influenced the changes in muscle mass observed in this study, there was no significant difference in the frequency of severe esophagitis or loss of appetite between the muscle maintenance and loss groups. Therefore, these adverse events are unlikely to have had a substantial impact on muscle mass.

This study had several limitations. First, this was a small retrospective study limited to a Japanese population, so selection bias could not be eliminated. Secondly, we were unable to collect data on inflammatory markers, which are necessary to substantiate the relationship between inflammation and muscle mass. Third, data on several adverse events during CCRT were not collected; therefore, we could not assess the effects of CCRT adverse events on efficacy. Fourth, information on smoking history, which could influence the efficacy of immunotherapy, was lacking. Therefore, it is possible that the muscle maintenance group had a higher proportion of patients with a history of smoking [47]. Fifth, comprehensive genomic data, including on TMB, were lacking. These data, which were not measured in this study, may have influenced the therapeutic effect of durvalumab. Sixth, because we did not investigate the rehabilitation interventions during the entire treatment period, we could not confirm the relationship between exercise and muscle mass. Lastly, there may have been potential confounding factors, such as patient comorbidities and differences in treatments between facilities.

## 5. Conclusions

Evaluating longitudinal muscle loss, rather than sarcopenia before treatment alone, may predict the efficacy of durvalumab maintenance therapy following concurrent chemoradiotherapy in patients with NSCLC better. However, this was a small retrospective study, and further prospective studies are required to confirm these findings.

## Figures and Tables

**Figure 1 cancers-16-03037-f001:**
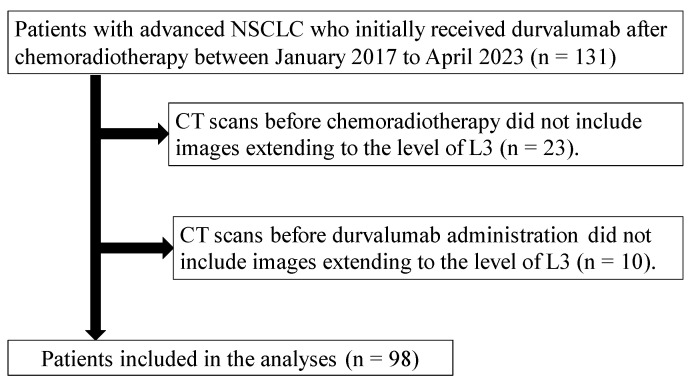
Flow diagram of patient enrollment. A total of 131 patients with advanced NSCLC received at least one cycle of durvalumab treatment after concurrent chemoradiotherapy between January 2017 and April 2023 at four medical institutions (*n* = 131). We excluded cases for the following reasons: the CT images before chemoradiotherapy did not include images extending to the L3 level (*n* = 23) and the CT images before durvalumab administration did not include images extending to the L3 level (*n* = 10). Ultimately, 98 patients were included in the analysis. NSCLC, non-small cell lung cancer; CT, computed tomography; L3, third lumbar vertebra.

**Figure 2 cancers-16-03037-f002:**
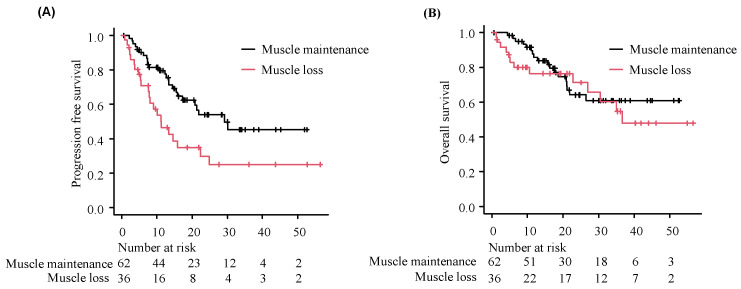
Comparison of the treatment effect of durvalumab between the muscle maintenance and loss groups. We compared the PFS and OS between the muscle maintenance and loss groups; Kaplan–Meier curves are shown for PFS (**A**) and OS (**B**). PFS: progression-free survival; OS: overall survival.

**Table 1 cancers-16-03037-t001:** Patient characteristics.

Characteristic *n* (%)	Total *N* = 98(%)	Muscle Maintenance Group *N* = 62(%)	Muscle Loss Group *N* = 36(%)	*p*-Value
Age (years) Median [range] <75 ≥75	71.5 [49–87] 60 (61.2) 38 (38.8)	70.5 [49–87] 40 (64.5) 22 (35.5)	72.0 [49–84] 20 (55.6) 16 (44.4)	0.55 0.40
Gender Male Female	73 (74.5) 25 (25.5)	46 (74.2) 16 (25.8)	27 (75.0) 9 (25.0)	1.00
ECOG-PS 0 1	41 (41.8) 57 (58.2)	25 (40.3) 37 (59.7)	16 (44.4) 20 (55.6)	0.83
Stage IIB IIIA IIIB IIIC Postoperative recurrence	4 (4.1) 35 (35.7) 39 (39.8) 14 (14.3) 6 (6.1)	1 (1.6) 19 (30.6) 27 (43.5) 10 (16.1) 5 (8.1)	3 (8.3) 16 (44.4) 12 (33.3) 4 (11.1) 1 (2.8)	0.24
Histology Adeno Others	45 (45.9) 53 (54.1)	25 (40.3) 37 (59.7)	20 (55.6) 16 (44.4)	0.21
Driver oncogenes EGFR mutation ALK rearrangement Others Negative Unknown	8 (8.2) 1 (1.0) 5 (5.1) 71 (72.4) 14 (13.3)	4 (6.5) 1 (1.6) 4 (6.5) 45 (72.6) 9 (12.9)	4 (11.1) 0 (0.0) 1 (2.8) 26 (72.2) 5 (13.9)	0.86
PD-L1 <50% ≥50% Unknown	47 (46.9) 39 (40.8) 12 (12.2)	28 (45.2) 27 (43.5) 7 (11.3)	19 (52.8) 12 (33.3) 5 (13.9)	0.62
BMI-kg/m^2^ (pre-chemoradiotherapy) Median [range] <22 ≥22	21.9 [15.6–30.5] 52 (53.1) 46 (46.9)	21.5 [16.5–28.3] 38 (61.3) 24 (38.7)	23.0 [15.6–30.5] 14 (38.9) 22 (61.1)	0.005
BMI -kg/m^2^ (post-chemoradiotherapy) Median [range] <22 ≥22	21.2 [15.2–28.7] 63 (64.3) 35 (35.7)	21.0 [15.4–28.7] 44 (71.0) 18 (29.0)	21.8 [15.2–27.6] 19 (52.8) 17 (47.2)	0.08
PMI-cm^2^/m^2^ (pre-chemoradiotherapy) Median [range] Male Female	5.03 [2.24–10.25] 3.52 [2.23–6.50]	4.68 [2.24–10.25] 3.28 [2.23–6.14]	5.25 [3.55–8.22] 4.32 [2.77–6.50]	0.01 0.07
PMI-cm^2^/m^2^ (post-chemoradiotherapy) Median [range] Male Female	4.60 [2.31–9.58] 3.21 [2.09–6.28]	4.66 [2.31–9.58] 3.16 [2.41–6.28]	4.43 [2.97–6.47] 3.58 [2.09–4.38]	0.61 0.89
Sarcopenia (pre-chemoradiotherapy) yes no	82 (83.7) 16 (16.3)	54 (87.1) 8 (12.9)	28 (77.8) 8 (22.2)	0.26
Sarcopenia (post-chemoradiotherapy) yes no	87 (88.8) 11 (11.2)	54 (87.1) 8 (12.9)	33 (91.7) 3 (8.3)	0.74
Cachexia yes no	53 (54.1) 45 (45.9)	32 (51.6) 30 (48.4)	21 (58.3) 17 (41.7)	0.54
Period between CCRT and ICIs (days) Median [range]	15.0 [1.0–49.0]	12.0 [1.0–45.0]	21.5 [1.0–49.0]	0.002
Chemotherapy CDDP+DTX CBDCA+PTX Daily CBDCA CDDP+VNR Weekly CBDCA+PTX Others	24 (24.5) 45 (45.9) 7 (7.1) 7 (7.1) 13 (13.3) 2 (2.0)	17 (27.4) 26 (41.9) 5 (8.1) 4 (6.5) 9 (14.5) 1 (2.8)	7 (19.4) 19 (52.8) 2 (5.6) 3 (8.3) 4 (11.1) 1 (1.6)	0.87
ICI completion rate	49 (50.0)	36 (58.1)	13 (36.1)	0.07
**Adverse Events during Concurrent Chemoradiotherapy**				
Leukopenia Any grade Grade 3 or more	49 (50) 40 (40.8)	34 (54.8) 28 (45.2)	15 (41.7) 12 (33.3)	0.30
Neutropenia Any grade Grade 3 or more	43 (43.9) 35 (35.7)	29 (46.8) 24 (38.7)	14 (38.9) 11 (30.6)	0.53
Thrombocytopenia Any grade Grade 3 or more	20 (20.4) 0	12 (21.0) 0	8 (22.2) 0	1.00
Febrile neutropenia Grade 3 or more	6 (6.1)	4 (6.5)	2 (5.6)	1.00
Anemia Any grade Grade 3 or more	25 (25.5) 3 (3.1)	16 (25.8) 0	9 (25.0) 3 (8.3)	1.00
Radiation pneumonitis Any grade Grade 3 or more	42 (42.9) 2 (2.0)	27 (43.5) 1 (1.6)	15 (41.7) 1 (2.8)	1.00
Anorexia Any grade Grade 3 or more	34 (34.7) 11 (11.2)	22 (35.5) 8 (12.9)	12 (33.3) 3 (8.3)	1.00
Radiation esophagitis Any grade Grade 3 or more	41 (41.8) 14 (14.3)	26 (41.9) 8 (12.9)	15 (41.7) 6 (16.7)	1.00
Hypoalbuminemia Any grade Grade 3 or more	18 (18.4) 0	12 (19.4) 0	6 (16.7) 0	0.79

Abbreviation; ECOG-PS: Eastern Cooperative Group performance status; Adeno: adenocarcinoma; EGFR: epidermal growth factor receptor; ALK: anaplastic lymphoma kinase; PD-L1: programmed cell death ligand 1; CDDP: cisplatin; DTX: docetaxel; CBDCA: carboplatin; PTX: paclitaxel; VNR: vinorelbine; BMI: body mass index; PMI: psoas muscle index; CCRT: concurrent chemoradiotherapy; ICI: immune checkpoint inhibitor.

**Table 2 cancers-16-03037-t002:** Univariable and multivariable analyses of PFS.

Predictors of PFS	Crude HR	95% CI	*p*-Value	Adjusted HR	95 CI	*p*-Value
Muscle loss rate-% (<10 vs. ≥10)	0.47	0.26–0.83	*p* < 0.05	0.478	0.26–0.86	*p* < 0.05
Age (years) (<75 vs. ≥75)	0.54	0.31–0.97	*p* < 0.05	0.59	0.33–1.07	0.08
ECOG-PS (0 vs. 1)	1.11	0.62–1.97	0.73	1.28	0.60–1.97	0.78
Histology (Adeno vs. others)	0.69	0.38–1.23	0.21	NI	NI	NI
Mutation (+ vs. −)	0.83	0.35–1.95	0.67	NI	NI	NI
PD-L1 (≥50% vs <50%)	1.06	0.70–1.63	0.78	1.18	0.64–2.18	0.59
Pre-chemotherapy BMI (kg/m^2^) (≥22 vs. <22)	1.18	0.67–2.10	0.57	NI	NI	NI
Post-chemotherapy BMI (kg/m^2^) (≥22 vs. <22)	0.77	0.41–1.43	0.40	NI	NI	NI
Cachexia (No vs. Yes)	0.68	0.38–1.22	0.19	NI	NI	NI
Sarcopenia (pre-chemotherapy) (No vs. Yes)	0.44	0.16–1.22	0.11	NI	NI	NI
Sarcopenia (post-chemotherapy) (No vs. Yes)	0.32	0.08–1.33	0.12	NI	NI	NI

Multivariable analysis adjusted for age, ECOG-PS score, and PD-L1 expression. Abbreviations: PFS, progression-free survival; HR, hazard ratio; CI, confidence interval; ECOG-PS, Eastern Cooperative Group performance status; Adeno, adenocarcinoma; PD-L1, programmed cell death ligand 1; BMI, body mass index.

## Data Availability

The datasets used and analyzed during the current study are available from the corresponding author upon reasonable request.

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
