# Peer review of "The Significance of Longitudinal Psoas Muscle Loss in Predicting the Maintenance Efficacy of Durvalumab Treatment Following Concurrent Chemoradiotherapy in Patients with Non-Small Cell Lung Cancer: A Retrospective Study"

_cancers, 2024, doi:10.3390/cancers16173037_

Round 1

Reviewer 1 Report

Comments and Suggestions for Authors

This is a very interesting paper that evaluated the longitudinal muscle loss in predicting the maintenance efficacy of durvalumab treatment following con-3 current chemoradiotherapy in patients with NSCLC in a retrospective cohort of 89 patients.  This study could have important implications and potentially provide interventions/instructions on maintaining weight in patients with NSCLC. However, there are several minor points that need to be addressed. 

1) More comprehensive discussion and relevant references should be included in the introduction,

Line 54-57, In terms of molecular biomarkers for ICI, actually there have been a couple of research looking into the biomarkers (e.g., TMB and aneuploidy) predicting ICI response and outcomes in addition to PD-L1 TPS, in particular in the setting of chemoradiation therapy. 

Ricciuti B, Wang X, Alessi JV, et al. Association of High Tumor Mutation Burden in Non–Small Cell Lung Cancers With Increased Immune Infiltration and Improved Clinical Outcomes of PD-L1 Blockade Across PD-L1 Expression Levels. JAMA Oncol. 2022;8(8):1160–1168. doi:10.1001/jamaoncol.2022.1981 

Spurr, L.F., Pitroda, S.P. Exploiting tumor aneuploidy as a biomarker and therapeutic target in patients treated with immune checkpoint blockade. npj Precis. Onc. 8, 1 (2024). https://doi.org/10.1038/s41698-023-00492-8

2) Definition of muscle maintenance group lacks detailed explanation. The authors chose 10% muscle in the lumbar skeletal muscle area to define the two groups. More references and rationale are needed to justify and explain the decision to the readers. 

- Regarding 10%, it will be helpful that the authors could do some sensitivity analyses using different cutoffs to show whether the results are consistent. 

- Regarding the muscle area, it will be helpful to discuss more on this selection. There has been a lot of discussion on characterizing body composition using CT scans. Some groups used T1-T12 while others used T4. If the authors are specifically using lumbar skeletal muscle area, then probably it is better to make it clear in the title. 

- The author didn’t include smoking history in this work. I wonder if they can include this important information as it may be correlated with muscle maintenance vs. loss group at baseline and therefore could potentially bias the results. They should also mention this as a limitation. 

- The multivariate analyses throughout the manuscript should be corrected with multivariable analyses. 

- In the discussion, the authors should also mention that a lack of comprehensive genomic data including TMB could be a limitation. 

Comments on the Quality of English Language

No big issues in the language. 

Author Response

Comments1: More comprehensive discussion and relevant references should be included in the introduction,

Line 54-57, In terms of molecular biomarkers for ICI, actually there have been a couple of research looking into the biomarkers (e.g., TMB and aneuploidy) predicting ICI response and outcomes in addition to PD-L1 TPS, in particular in the setting of chemoradiation therapy. 

Ricciuti B, Wang X, Alessi JV, et al. Association of High Tumor Mutation Burden in Non–Small Cell Lung Cancers With Increased Immune Infiltration and Improved Clinical Outcomes of PD-L1 Blockade Across PD-L1 Expression Levels. JAMA Oncol. 2022;8(8):1160–1168. doi:10.1001/jamaoncol.2022.1981 

Spurr, L.F., Pitroda, S.P. Exploiting tumor aneuploidy as a biomarker and therapeutic target in patients treated with immune checkpoint blockade. npj Precis. Onc. 8, 1 (2024). https://doi.org/10.1038/s41698-023-00492-8

Response1: Thank you for your suggestions. I have added some additional references, including information on ICI biomarkers and the references provided by the reviewer (lines 55-66).

Comments2: Definition of muscle maintenance group lacks detailed explanation. The authors chose 10% muscle in the lumbar skeletal muscle area to define the two groups. More references and rationale are needed to justify and explain the decision to the readers. 

  • Regarding 10%, it will be helpful that the authors could do some sensitivity analyses using different cutoffs to show whether the results are consistent

Response2: In response to your suggestion, we have added additional references, alongside our own study, to support the use of a 10% cutoff. Additionally, we utilized an ROC curve to validate that a 10% reduction in muscle mass is an appropriate threshold, and we have included the results in the manuscript (line 113-119).  

Comments3: Regarding the muscle area, it will be helpful to discuss more on this selection. There has been a lot of discussion on characterizing body composition using CT scans. Some groups used T1-T12 while others used T4. If the authors are specifically using lumbar skeletal muscle area, then probably it is better to make it clear in the title.

Response3: Following your suggestion, we change the title “Significance of longitudinal psoas muscle loss in predicting the maintenance efficacy of durvalumab treatment following concurrent chemoradiotherapy in patients with NSCLC: a retrospective study”.

Comments4; The author didn’t include smoking history in this work. I wonder if they can include this important information as it may be correlated with muscle maintenance vs. loss group at baseline and therefore could potentially bias the results. They should also mention this as a limitation. 

Response4; As per your suggestion, we have added to limitation that we do not collect data on smoking history. (line 315-316)

Comments5;The multivariate analyses throughout the manuscript should be corrected with multivariable analyses. 

Response5; As per your suggestion, we have changed all instances of "multivariate" to "multivariable."

Comments6; In the discussion, the authors should also mention that a lack of comprehensive genomic data including TMB could be a limitation. 

Response6; In response to your comment, we have noted in the text that the lack of comprehensive genomic data, including TMB, represents a limitation of our study (line 318-319).

Reviewer 2 Report

Comments and Suggestions for Authors

The introduction is well-structured, providing a comprehensive overview of the current understanding of immune checkpoint inhibitors (ICIs) in NSCLC and the potential role of sarcopenia as a predictor of treatment response. However, the study does not introduce any novel hypotheses or innovative methodologies. The focus on sarcopenia and muscle loss as predictors of response to durvalumab is consistent with existing literature, which limits the novelty of the work.

The methods section is detailed, with clear descriptions of patient inclusion and exclusion criteria, data collection methods, and imaging techniques used to assess sarcopenia. The choice of parameters, such as lumbar skeletal muscle area and psoas muscle mass index (PMI), is appropriate and aligns with prior studies. However, the lack of details regarding potential confounders, such as patient comorbidities and variations in treatment between centers, may introduce bias into the study's findings. Furthermore, the methodology used is not innovative, as it follows established protocols for assessing sarcopenia using CT imaging.

The statistical analyses are adequate, with Kaplan-Meier survival curves used for progression-free survival (PFS) and overall survival (OS), and Cox proportional hazards models employed to identify predictors of PFS. The use of EZR software, a graphical interface for R, is appropriate for biostatistical analysis. However, the small sample size and retrospective nature of the study limit its statistical power and increase the risk of selection and information bias. The study does not introduce any new statistical techniques or approaches, further emphasizing the lack of innovation.

The results are presented clearly and consistently with the study’s hypotheses and methodologies. The study found that patients who maintained muscle mass during chemoradiotherapy had significantly longer PFS compared to those who experienced significant muscle loss. However, there was no significant difference in OS between these groups. The lack of impact on OS, despite the significant difference in PFS, raises questions about the overall clinical relevance of muscle loss as a predictor of long-term outcomes. The results support the existing body of literature on the importance of muscle maintenance in cancer treatment but do not provide new insights or breakthroughs.

The discussion is thorough, with a critical analysis of the results and the study’s limitations. The authors propose plausible mechanisms for the observed association between muscle loss and reduced efficacy of durvalumab, including chronic inflammation and the potential benefits of exercise. However, the discussion does not go beyond existing theories and fails to explore innovative angles or novel hypotheses. The authors acknowledge the limitations of their study, including its retrospective design, small sample size, and potential for unmeasured confounding factors. These limitations further highlight the lack of originality and the need for more robust prospective studies.

The manuscript is consistent overall, with a well-structured argument linking the introduction, methods, results, and discussion. However, the study does not introduce any new concepts, methods, or findings that would significantly advance the field. The conclusions are supported by the data presented, but they align closely with what is already known about sarcopenia and cancer treatment outcomes.

The study presents several potential biases:

  • Selection Bias: The retrospective design and inclusion of only patients who completed at least one cycle of durvalumab may introduce selection bias.
  • Survivorship Bias: The study may be influenced by survivorship bias, as only those who survived long enough to receive durvalumab were included.
  • Confounding Bias: Potential confounding factors, such as comorbidities and differences in treatment regimens between centers, were not fully accounted for.
  • Information Bias: The reliance on CT imaging at specific time points for assessing sarcopenia may lead to information bias if the imaging was not consistent across all patients.
  • Generalizability Bias: The study's findings may not be generalizable to other populations, as it was conducted in a specific demographic and geographic context (Japanese population).

A review of recent literature on the association between sarcopenia and ICI therapy in NSCLC patients confirms that this is a well-established area of research. The findings of this study are consistent with previous research, which shows that sarcopenia is associated with worse outcomes in cancer patients. However, the study does not contribute new knowledge or innovative approaches to the field. This lack of innovation limits the impact of the study on advancing the understanding of sarcopenia in cancer treatment.

While the study is methodologically sound and contributes to the ongoing discussion of sarcopenia as a predictor of treatment outcomes in NSCLC patients, it does not offer new insights or significant advancements in the field. The findings are consistent with existing literature, and the study lacks the innovation necessary to make a substantial impact. 

Author Response

Comments1: The introduction is well-structured, providing a comprehensive overview of the current understanding of immune checkpoint inhibitors (ICIs) in NSCLC and the potential role of sarcopenia as a predictor of treatment response. However, the study does not introduce any novel hypotheses or innovative methodologies. The focus on sarcopenia and muscle loss as predictors of response to durvalumab is consistent with existing literature, which limits the novelty of the work.

Response1: Thank you for highlighting these important points and for your thorough review.

Comments2: The methods section is detailed, with clear descriptions of patient inclusion and exclusion criteria, data collection methods, and imaging techniques used to assess sarcopenia. The choice of parameters, such as lumbar skeletal muscle area and psoas muscle mass index (PMI), is appropriate and aligns with prior studies. However, the lack of details regarding potential confounders, such as patient comorbidities and variations in treatment between centers, may introduce bias into the study's findings. Furthermore, the methodology used is not innovative, as it follows established protocols for assessing sarcopenia using CT imaging.

Response2; As you pointed out, our study lacks detailed information on potential confounding factors, such as patient comorbidities and variations in treatment across facilities, which may introduce bias into our findings. We have added these points as limitations of our study. (line 322-323)

Comments3: The statistical analyses are adequate, with Kaplan-Meier survival curves used for progression-free survival (PFS) and overall survival (OS), and Cox proportional hazards models employed to identify predictors of PFS. The use of EZR software, a graphical interface for R, is appropriate for biostatistical analysis. However, the small sample size and retrospective nature of the study limit its statistical power and increase the risk of selection and information bias. The study does not introduce any new statistical techniques or approaches, further emphasizing the lack of innovation.

Response3: As you pointed out, this study is a retrospective analysis, and the small sample size is considered one of the limitations of the study, as noted in the manuscript. (line 311-312)

Comments4: The results are presented clearly and consistently with the study’s hypotheses and methodologies. The study found that patients who maintained muscle mass during chemoradiotherapy had significantly longer PFS compared to those who experienced significant muscle loss. However, there was no significant difference in OS between these groups. The lack of impact on OS, despite the significant difference in PFS, raises questions about the overall clinical relevance of muscle loss as a predictor of long-term outcomes. The results support the existing body of literature on the importance of muscle maintenance in cancer treatment but do not provide new insights or breakthroughs.

Response4; As you pointed out, this study found that while the maintenance of muscle mass during chemoradiotherapy was a predictor of PFS in the context of ICI treatment, it did not show a significant association with OS. It is known that the prognosis of locally advanced lung cancer is influenced by various factors beyond systemic conditions, such as muscle mass, including subsequent recurrence and responsiveness to further treatments, which may explain the study's findings. Additionally, the small cohort size in this analysis may have resulted in insufficient power to detect an impact on OS. The limitation of the study being a small cohort is acknowledged in the manuscript.

Comments5: The discussion is thorough, with a critical analysis of the results and the study’s limitations. The authors propose plausible mechanisms for the observed association between muscle loss and reduced efficacy of durvalumab, including chronic inflammation and the potential benefits of exercise. However, the discussion does not go beyond existing theories and fails to explore innovative angles or novel hypotheses. The authors acknowledge the limitations of their study, including its retrospective design, small sample size, and potential for unmeasured confounding factors. These limitations further highlight the lack of originality and the need for more robust prospective studies.

The manuscript is consistent overall, with a well-structured argument linking the introduction, methods, results, and discussion. However, the study does not introduce any new concepts, methods, or findings that would significantly advance the field. The conclusions are supported by the data presented, but they align closely with what is already known about sarcopenia and cancer treatment outcomes.

The study presents several potential biases:

  • Selection Bias: The retrospective design and inclusion of only patients who completed at least one cycle of durvalumab may introduce selection bias.
  • Survivorship Bias: The study may be influenced by survivorship bias, as only those who survived long enough to receive durvalumab were included.
  • Confounding Bias: Potential confounding factors, such as comorbidities and differences in treatment regimens between centers, were not fully accounted for.
  • Information Bias: The reliance on CT imaging at specific time points for assessing sarcopenia may lead to information bias if the imaging was not consistent across all patients.
  • Generalizability Bias: The study's findings may not be generalizable to other populations, as it was conducted in a specific demographic and geographic context (Japanese population).

Response5; We have added several potential research biases that were not specified in the study to the limitations section.

Comments6; A review of recent literature on the association between sarcopenia and ICI therapy in NSCLC patients confirms that this is a well-established area of research. The findings of this study are consistent with previous research, which shows that sarcopenia is associated with worse outcomes in cancer patients. However, the study does not contribute new knowledge or innovative approaches to the field. This lack of innovation limits the impact of the study on advancing the understanding of sarcopenia in cancer treatment.

Response6; As noted, the findings of this study are limited; however, they are expected to serve as a foundation for future prospective research, as stated in the manuscript. (line 328-329)

Comments7; While the study is methodologically sound and contributes to the ongoing discussion of sarcopenia as a predictor of treatment outcomes in NSCLC patients, it does not offer new insights or significant advancements in the field. The findings are consistent with existing literature, and the study lacks the innovation necessary to make a substantial impact. 

Response7; As noted, the findings of this study are limited; however, this is the first report on the analysis of muscle mass at multiple points in this cohort and its subsequent impact on ICI treatment efficacy. Given that this is a small cohort study, it is expected to serve as a foundation for future prospective research, as stated in the manuscript. (lines 328-329).